# OpenReview forum: "Cyber-Zero: Training Cybersecurity Agents without Runtime"
_ICLR.cc/2026/Conference — ICLR 2026 Poster_

### Official Review · Reviewer_3STn · 2025-10-30

**Soundness:** 3
**Presentation:** 4
**Contribution:** 3
**Rating:** 8
**Confidence:** 3

**Summary:**

*Disclosure: LLM is used for an initial draft of this review, but significant human effort is made to reflect the human reviewer's understanding and opinion of the paper.*

This paper addresses a current bottleneck in training large language model (LLM) agents for cybersecurity: the scarcity of high-quality, agentic training data. The authors note that unlike software engineering (SWE) where ample executable runtime environments are available to generate trajectories (e.g., from GitHub issues), cybersecurity environments are often ephemeral, restricted, or unavailable. This data scarcity has led to a significant performance gap between proprietary models and open-weight models.

To solve this, the paper introduces CYBER-ZERO, a novel and practical framework for synthesizing high-quality agent trajectories without needing an executable runtime. The core idea is to leverage publicly available Capture The Flag (CTF) writeups and simulate runtime environment there upon with a LLM. Specifically, CYBER-ZERO uses a "persona-driven dual-LLM" simulation to reverse-engineer the runtime behavior described in these writeups:
1. A "CTF Player" LLM (which does not see the solution) attempts to solve the challenge from scratch.
2. A "Bash Terminal" LLM (which does see the writeup and flag) acts as a "weak oracle," simulating realistic terminal outputs and providing minimal hints when the Player model gets stuck.

The result is a large-scale dataset (over 9,400 trajectories) that captures realistic, long-horizon interactions, including debugging and exploration. The authors use this dataset to fine-tune open-weight LLMs.

Models trained with CYBER-ZERO show significant performance gains (up to 13.1% absolute improvement) on three prominent CTF benchmarks. The best-trained open model, CYBER-ZERO-32B, achieves new state-of-the-art performance among open-weight models, matching more advanced models like Claude-3.5-Sonnet and DeepSeek-V3-0324.

**Strengths:**

- The paper tackles a well-defined, significant, and difficult problem: the data-scarcity bottleneck for training open-source cybersecurity agents. The proposed runtime-free framework, CYBER-ZERO, is a highly novel and pragmatic solution that effectively "upcycles" existing natural-language writeups into valuable, structured training data.
- The presented results are strong and convincing. The performance gains are clear and substantial across multiple models and benchmarks. The results model match the performance of top proprietary models while being far more cost-effective. Throughout ablations are also presented.
- The writing is clear and engaging.

**Weaknesses:**

- The quality of the generated traces depend greatly on the simulator LLM: it may need to, for example, simulate decompilation results and kernel exploits. Simulating these itself requires significant cybersecurity-related capabilities (not necessarily on the attack side, though), especially when the traces deviate from the writeups. While the shown experiment results are convincing, this may limit the upper bound of the method, as inaccurate traces may be less helpful for training.
- The authors state the hint mechanism is crucial for data collection, which may lead to the generated trajectories being guided by the writeups rather than authentic, self-driven exploration. This may limit the result LLMs' capability on discovering truly novel exploits.

**Questions:**

- Is the player LLM given the content of the writeup? The main body seems to suggest that it is not provided to the LLM, but the prompt provided in appendix B.2 mentioned the use of writeup.
- The performance of DeepSeek-V3-0324 on NYU CTF seems low compared to closed-weight systems. Curious if you have explanation on that.

---

> ### Author Response · Authors · 2025-11-23
>
> Thank you for recognizing our work!
>
> > The authors state the hint mechanism is crucial for data collection, which may lead to the generated trajectories being guided by the writeups rather than authentic, self-driven exploration.
>
> Please refer to our updated content on Lines 234-235. Specifically, we did remove the hints from the collect trajectories by searching for the hint tags before training, where the hints won’t be seen from the fine-tuned models anymore. In addition, the hints tend to only guide the player model to rediscover anything it may overlook instead of directly telling any solutions.
>
> > The prompt provided in appendix B.2 mentioned the use of writeup.
>
> The player model is never given the writeup. However, as the terminal model can access the writeup, there is still a chance that the hint it provides can contain the phrases like “the writeup suggests …” even when it is asked not to. Therefore, we still ask the player model to never mention the words like “writeup” when it sees them inside the hints.
>
> > The performance of DeepSeek-V3-0324 on NYU CTF seems low compared to closed-weight systems.
>
> Based on the observations, we found that DeepSeek-V3-0324 is still better at easier tasks, which may explain why it achieved comparable performance to other closed models. While Cybench is supposed to be harder, it has only 40 challenges compared to 91 in InterCode-CTF and 192 in NYU CTF Bench. It’s relatively difficult to demonstrate the performance difference.

---

### Official Review · Reviewer_u3TU · 2025-10-30

**Soundness:** 3
**Presentation:** 3
**Contribution:** 3
**Rating:** 8
**Confidence:** 4

**Summary:**

This paper presents a novel approach of training LMs to be better Cybersecurity agents based on synthetic training trajectories. These training trajectories are generated using persona-driven LMs simulation without a real execution environment, i.e., one LM/agents impersonates the execution environment and simulates execution outputs, another LM/agent impersonates the agent/player.

Experiments are performed based on the Qwen family, showing large gains on 8B, 14B, and 32B models.
Analysis parts look at scaling with number of training trajectories and multiple rollouts (best@k).

**Strengths:**

Core contribution:

* Good motivation for why this work is important (scarcity of high-quality training data for cybersec applications, because execution environments are harder to obtain).
* Very clearly data synthesis methodology which seems to be original. The data synthesis strategy seems to be very easy and applicable.
* The training dataset itself will also be a good asset and seems to be larger than previously available datasets. Clear demonstration of effectiveness of method, with large gains over the baseline models.
* Good analysis sections, looking at pass@k and also at scaling with number of training trajectories
* Very smooth read.

Additional contributions:

* EnIGMA+ agent scaffolding improvements show good engineering and will help others use the scaffold.
* Identification of broken benchmark instances also good contribution.

**Weaknesses:**

Minor comment:

The abstract states "Our best model, CYBER-ZERO-32B, [matches] the capabilities of proprietary systems like DeepSeek-V3-0324 and Claude-3.5-Sonnet", this seems to be an overstatement if one considers Fig 1 or Fig 3.

**Questions:**

What exact steps were taken to avoid contamination? The new dataset spans more than 1k challenges. The paper only says "manually removed 47 overlapping challenges", but doesn't go into details how these were identified. Since this is a crucial point for the validity of the results, some additional details or studies would be great to see (e.g., looking for overlaps in flag strings, or specific n-grams etc.).

Presentational things I noticed/don't understand

* Fig 2: I am still confused by the arrows in the upper part of the figure. There's two personas, so what's the "LM" doing in the middle?  Or does this mean that the CTF challenge text is created based on the writeup with the help of the LM?

---

> ### Author Response · Authors · 2025-11-23
>
> Thank you for recognizing our work.
>
> > What exact steps were taken to avoid contamination?
>
> We note that we are able to collect the metadata information (e.g., competition name, hosted year, challenge name) of the tasks from both our benchmark suite and the ones collected from CTFtime. Therefore, it is quite straightforward to remove the challenges used for training purposes by mapping each challenge to the concatenated string of competition name,  hosted year, and challenge name. After applying this rule-based decontamination process, we further manually checked the metadata of the remaining CTF challenges against the benchmarks.
>
>
> > I am still confused by the arrows in the upper part of the figure. There's two personas, so what's the "LM" doing in the middle? Or does this mean that the CTF challenge text is created based on the writeup with the help of the LM?
>
> The CTF challenge text is indeed created based on the writeup with the help of the LM. We documented this part on Lines 156-158.

---

### Official Review · Reviewer_1n45 · 2025-10-30

**Soundness:** 3
**Presentation:** 3
**Contribution:** 4
**Rating:** 6
**Confidence:** 5

**Summary:**

This paper presents Cyber-Zero, a novel training data generation method that uses persona-based LLMs to produce realistic multi-turn trajectories of CTF task solving from human writeups without involving runtimes or actually executing commands. Cyber-Zero performs supervised  finetuning of open-source Qwen models and obtains noticeable percentage improvement, demonstrating that the generated trajectories are useful to improve cybersecurity performance of LLMs.

**Strengths:**

- The runtime-free framework to synthesize multi-turn trajectories from CTF writeups is an impactful technical contribution
- The paper provides comprehensive evaluation of multiple commercial and open-source LLMs on CTF benchmarks, and how their fine-tuning improves performance. Specifically, the result of SWE-Agent-LM not generalizing to CTF tasks is interesting
- The rectification of problematic challenges of existing CTF benchmarks is appreciated

**Weaknesses:**

- Aspects of the main contribution of synthesizing multi-turn trajectories need to be elaborated, such as how and when are hints provided. Please see "Questions" for specifics
- The paper lacks evaluation of whether this finetuning method has trained an LLM that overfits to a specific agentic format like EnIGMA. It raises pertinent questions about how the trained LLM would perform with a different agentic framework that is also targeted towards CTF tasks.
- Some aggregate statistics should be provided for the CTF writeups collected from CTFtime, to understand the composition of the finetuning data
- The contribution of EnIGMA+ seems insignificant and superfluous as it is not adequately discussed in the text, and it seems like a simple implementation change of deduplicating the docker port numbers

**Questions:**

- Table 1: It seems unnecessary to compare and contrast CVE-Bench, NYU CTF Bench, Cybench with Cyber-Zero as the former are benchmarks to evaluate LLM agents on cybersecurity tasks, while the later is a dataset for fine-tuning
- While Figure 2 gives an example of how synthetic trajectories are generated, some more details will be helpful to clarify the entire picture
	- How and when does the terminal model decide to provide hints to the player model?
	- How does the terminal model decide what hints to give?
	- If there is malformed, hallucinated, or unstructured outputs by either models, how is it handled?
- Line 144: provide text links of "CTFTime" and "CTF Archives" as footnotes instead of linking in the PDF
- Line 151: How does the DeepSeek model obtain any data that is missing from the CTF writeups, such as file contents?
- Line 247: Please provide details of the binary filter

---

> ### Author Response · Authors · 2025-11-23
>
> Thank you very much for your thorough and thoughtful review. We greatly appreciate the time and effort you have dedicated to providing detailed feedback. Below, we address your questions point by point.
>
> > It seems unnecessary to compare and contrast CVE-Bench, NYU CTF Bench, Cybench.
>
> We appreciate your comment. However, we think that most researchers in the LLM domain are not very familiar with cybersecurity. By providing the comparisons of cybersecurity datasets (regardless of training and evaluation), the researchers may have a better understanding of the current progress of this domain and our contributions in this paper.
>
> > How and when does the terminal model decide to provide hints to the player model?
>
> On Lines 211-213, we explain this process by “When the player makes repeated mistakes or takes an unproductive direction, the terminal model can inject minimal hints, marked with the special [HINT]...[/HINT] tags.” Specifically, we do not design or program a specific way to make the terminal model provide hints. We prompt the terminal model with “YOU **MUST NOT** HELP THE PLAYER SOLVE ANY STEPS OF THE CHALLENGE. IF THE PLAYER IS STUCK, YOU MUST PROVIDE HINTS AND REASONS BEHIND THE HINTS TO THE PLAYER IN THE FORMAT OF `[HINT] hint [/HINT]` WITH `---HINT_START---` AND `---HINT_END---`” inside the system prompt of the terminal model, as documented in Appendix B.3. Based on our observations from the generated data, the terminal model can reliably provide the hints only when the player model is off the track.
>
> > If there are malformed, hallucinated, or unstructured outputs by either model, how is it handled?
>
> As described on Lines 254-256, we designed the multi-layer validation to ensure data quality and consistency. Specifically, we mentioned that “we enforce format checks to verify Markdown consistency, structural alignment with the agent scaffold, and presence of a single command per player response” to avoid structurally malformed or unstructured outputs. For any hallucinated or low-quality outputs, we use the LLM-based binary filter (LLM-as-a-Judge) to assess the quality against the writeup context.
>
> > How does the DeepSeek model obtain any data that is missing from the CTF writeups, such as file contents?
>
> We note that this could be a misunderstanding of our data generation process. Our method does not rely on any real-world file content or runtime environment. The whole environment is fully simulated with the “world knowledge” of the terminal model. The only necessary information before trajectory generation is the synthetic CTF challenge metadata, which is illustrated in Figure 2 and documented on Lines 156-158.

---

> > ### Comment · Reviewer_1n45 · 2025-11-25
> > **Thank you for the clarifications, but please respond to the Weaknesses also**
> >
> > The concerns listed in "Questions" were all adequately answered. I appreciate the clarifications about the binary filter.
> >
> > However, most of the concerns listed in "Weaknesses" have not been addressed.

---

> > > ### Author Response · Authors · 2025-11-26
> > >
> > > Thank you for responding! Here are our responses to the remaining weaknesses:
> > >
> > > > The paper lacks evaluation of whether this finetuning method has trained an LLM that overfits to a specific agentic format like EnIGMA.
> > >
> > > We note that EnIGMA is the only framework targeting cybersecurity tasks (e.g., CTF) so far. Previous scaffoldings used in NYU CTF Bench and Cybench are very naive, without any tool interfaces specifically designed for security domains like metasploit and radare2. These frameworks are not considered as the specific scaffoldings in the orginal papers. We are happy to extend the experiments on other general cybersecurity agent scaffoldings when they are available.
> > >
> > > > Some aggregate statistics should be provided for the CTF writeups collected from CTFtime, to understand the composition of the finetuning data
> > >
> > > Please refer to the aggregate statistics originally mentioned on Lines 159-161. As we did not directly include any writeups during the fine-tuning stage, we believe that those aggregate statistics may hardly indicate the quality and structure of collected trajectories.
> > >
> > > > The contribution of EnIGMA+ seems insignificant and superfluous as it is not adequately discussed in the text, and it seems like a simple implementation change of deduplicating the docker port numbers
> > >
> > > The contribution of EnIGMA+ indeed lies more in the engineering perspective than technical novelty. We note the scaffolding changes including the dynamic docker port allocation are not straightforward and hard to explain without disclosing the details inside codebases. As this is a relatively minor contribution, we didn’t tend to describe the full implementation in the paper but rather making it publicly available for future usage.

---

### Official Review · Reviewer_B84o · 2025-10-31

**Soundness:** 3
**Presentation:** 3
**Contribution:** 1
**Rating:** 2
**Confidence:** 4

**Summary:**

This paper introduces Cyber-Zero, a framework for fine-tuning LLMs to perform better at CTFs. Specifically, they collect a large dataset of CTF writeups, synthetically generate trajectories, and then observe increased performance across three CTF datasets: InterCode, NYU, and Cybench.

**Strengths:**

The paper is well-organized and clear. The authors provide a helpful amount of detail and insight into their proposed method which cleverly uses a dual-LLM system to simulate agentic trajectories from previous CTF writeups that don't have accompanying runtime configurations.

**Weaknesses:**

This paper exclusively focuses on improving SOTA results of open source models on CTF benchmarks. The authors claim that Cyber Zero is a framework for training cybersecurity agents, when instead it is a framework for training CTF agents, just as ENIGMA is an agent scaffold strictly designed for CTFs and not for real world cybersecurity tasks. As a result, the usefulness of Cyber Zero is unproven. If the ENIGMA-scaffolded agents trained using Cyber Zero obtained state of the art results on more general benchmarks like MHBench, BountyBench, CVEBench, or XBOW's validation set, their claim of training cybersecurity agents would be more grounded.

Further, while model performance using Cyber Zero does improve upon the baseline of open source models, this method does not reach absolute state-of-the-art. I am unconvinced that the ends justify the means for anyone who might want to deploy such a system, particularly because it is only "proven" on CTFs.

The authors do not provide sufficient justification that their two-player synthetic generation pipeline is strictly required to achieve such results. For example, would single-shot trajectories yield similar results? And, relatedly, I am not convinced that simulating interactions in the fashion involved in this paper is a long-term worthwhile endeavor. Cybersecurity environments are no different from software engineering environments, and it is likelier that in the future environments with working runtimes that can scale will be ever-present.

Finally, the authors make claims that their method is much more cost-effective than using API-based models like Claude Sonnet 3.7 that are able to achieve state of the art results, and yet they do not include fine-tuning or data generation cost in this analysis.

**Questions:**

1. How much did data generation cost?
2. What was the total cost of fine-tuning these models on your hardware?
3. Why did you choose to focus strictly on CTFs for this paper?

---

> ### Author Response · Authors · 2025-11-23
>
> Thanks a lot for your valuable review. We would like to address your concerns as follows:
>
> > The paper is not evaluated on other “general” “real-world” cybersecurity benchmarks.
>
> Thanks for raising this concern. We **respectively argue** that our work targets **cross-domain** **offensive** cybersecurity tasks, as suggested on Lines 110-111. As mentioned on Lines 50-58, CTF serves as the standard to evaluate vulnerability exploitation. Even if the CTF challenges may not fully reflect the “real-world” scenarios, Cyber-Zero will help pave the way for future works in this direction by providing an open recipe for improving agentic cybersecurity capabilities.
>  First, there are limited well-established benchmarks focused on agentic cybersecurity capabilities. While we agree that the benchmarks you suggested can be useful, they are not being widely used to evaluate existing LLMs. As mentioned on Lines 56-58 and Lines 63-64, existing models from OpenAI, Anthropic, and xAI are all assessed on CTF benchmarks according to their model system cards. Second, the benchmarks you suggested are not general. Instead, these benchmarks lack cybersecurity domain diversity. For example, MHBench, CVE-Bench, and XBOW's validation benchmark are only designed for web applications. On the other hand, the CTF benchmark suite used in our paper is much more dynamic and hence more generic. As shown in Table 2, the evaluation suite covers 5 main domains including web applications and other miscellaneous tasks. Third, the recommended benchmarks are not large-scale enough to properly evaluate the model capabilities. MHBench only has 10 environments; BountyBench has 25 environments and 40 bugs; CVE-Bench has 40 environments. Compared to these small-scale benchmarks, our evaluation suite has 323 unique-environment-based tasks displayed in Table 2.
>
> > Cyber Zero does improve upon the baseline of open source models, this method does not reach absolute state-of-the-art.
>
> Thanks for the comment. We agree that “state-of-the-art performance” always changes, as the new models keep being developed. However, we’d like to highlight that Cyber-Zero is still the state-of-the-art open recipe to improve the models on agentic cybersecurity capabilities so far. In addition, we’ve toned down the words of “achieve state-of-the-art performance” in the paper.
>
> > The authors do not provide sufficient justification that their two-player synthetic generation pipeline is strictly required to achieve such results.
>
> Thanks for the feedback. We sincerely ask the reviewer to check Lines 187-191, where we discuss that “trajectories generated in a single pass are often overly linear and omit the trial-and-error, exploration, and self-repair behaviors commonly seen in human workflows (see Appendix G)”. More detailed results on single-shot trajectories can be viewed in Table 7.
>
> > Cybersecurity environments are no different from software engineering environments.
>
> We **respectively disagree** with this statement. As stated on Lines 322-323, we found that “SWE agents cannot generalize on cybersecurity tasks.” The previous state-of-the-art open-source SWE models, SWE-agent-LMs, are trained via SWE-agent, an agent scaffolding that shares a similar interface with EnIGMA and EnIGMA+, they demonstrate much worse performance on cybersecurity tasks. On Lines 376-377, we discussed that “these results underscore the importance of domain-specific training to enable effective generalization in agentic settings.”
>
> > The authors make claims that their method is much more cost-effective yet they do not include fine-tuning or data generation cost.
>
> We believe that this is a **misunderstanding** of our claim in the paper. On Lines 371-372, we clearly mentioned that “Fine-tuned agents achieve superior cost-effectiveness”, instead of suggesting that the Cyber-Zero method is cost-effective. Therefore, we only evaluate the inference cost instead of data generation and training cost. In addition, the cost comparison between training Cyber-Zero models and closed models like Claude Sonnet 3.7 is impossible, as we have no knowledge of developing these closed models. With this being said, we are still happy to provide some details on data generation: we hosted DeepSeek models on an Amazon EC2 p5 instance and completed the data generation within one day. The estimated cost is  31 * 24 =  744 USD. For training those Cyber-Zero models, we used at most 32 * 8 H100 GPUs and 5 hours per model. The maximum cost of training one model is 11.8 * 32 * 5 = 1888 USD.

---

### Author Response · Authors · 2025-11-23

We thank all reviewers for their detailed reviews. We have made the following updates to the paper with the additional blue text.

- **Replacement**: Correction to “state-of-the-art performance”, asked by Reviewer B84o.
- **Replacement**: Text links of "CTFTime" and "CTF Archives" as footnotes, asked by Reviewer 1n45.
- **Removal**: Overstatement in the abstract, asked by Reviewer 1n45.
- **Addition**: Descriptions of the binary filter mentioned in the policy improvement algorithm, asked by Reviewer 1n45.
- **Addition**: Detailed Post-processing of the collected trajectories, asked by Reviewer 3STn.

---

### Author Response · Authors · 2025-12-04
**Summary of the rebuttal**

Due to the rollback, we briefly summarize the rebuttal process here. All ongoing discussions occurred during the data leakage incident.

- **Reviewer B84o** who gave 2 **significantly misunderstood** the paper and raised various **invalid** questions. Most of the answers can be directly found in the orginal paper. The reviewer may lack a sense of responsibility and never replied to our responses.
- **Reviewer 1n45** who gave 6 agreed that we have adequately answered the questions. We have further responded to the remaining weaknesses as required.
- We also answered all the questions and concerns raised by **Reviewer u3TU** and **Reviewer 3STn**.

---

### Meta-Review · Area_Chair_RVFV · 2026-01-09

**Summary:**

This paper proposes the runtime-free framework, Cyber-Zero,  for synthesizing high-quality agent trajectories to train cybersecurity LLMs. As statemented in the manuscript, the proposed framework leverages publicly available CTF writeups and employs persona-driven LLM simulation to reverse-engineer runtime behaviors and generate realistic, long-horizon interaction sequences without actual environments. Overall, most of the initial reviews support that this manuscript should be accepted (B84o-rating-2, 1n45-rating-6, 3STn-rating-8, u3TU-rating-8). After carefully reading the concerns from B84o and the corresponding rebuttal from authors, it could be concluded that most of the issues are well addressed. This manuscript should be accepted.

**Reviewer Concerns:**

Basically, I think most of the concerns are well addressed. Some issues (question 1, 2, and 3) from B84o are not answered explicitly, while the correlated information are included in the rebuttal.

**Reviewer Scores:**

It seems 1n45 might improve his/her/they score but the B84o might not.

---

### Decision · Program_Chairs · 2026-01-26

Accept (Poster)